# Development and Validation of the War Worry Scale (WWS) in a Sample of Italian Young Adults: An Instrument to Assess Worry About War in Non-War-Torn Environments

**DOI:** 10.3390/ejihpe15020024

**Published:** 2025-02-09

**Authors:** Giorgio Maria Regnoli, Anna Parola, Barbara De Rosa

**Affiliations:** Department of Humanities, University of Naples Federico II, Via Porta di Massa 1, 80133 Naples, Italy; giorgiomaria.regnoli@unina.it (G.M.R.); anna.parola@unina.it (A.P.)

**Keywords:** worry about war, scale development, scale validation, psychological impact of war, young adults, mental health, exploratory factor analysis, confirmatory factor analysis

## Abstract

The expansion of wars around the world fosters a macrosocial stress with multilevel effects that also affect the mental health of populations not directly involved, in particular of evolutionary targets in delicate transition. The present study describes the process of development, validation, and evaluation of the psychometric properties of the War Worry Scale (WWS), an instrument that explores the psychological impact of war in contexts not directly involved and, in particular, in the target population of young Italian adults. The process of construct definition and item generation of the WWS is presented here and then verified in Study I, which, using a sample of 250 young adults (40.4% male and 59.6% female), describes the exploration of the factor structure of the instrument through exploratory factor analysis (EFA) and presents preliminary psychometric properties. An independent sample of 500 young adults (39.4% male; 60.6% female) was recruited for Study II, which describes the results of confirmatory factor analysis (CFA) supporting the second-order structure with two first-order dimensions, Worry about the Present (WWP) and Worry about the Future (WWF), composed of 10 items (5 per dimension). The internal consistency of the WWS, convergent, discriminant, and concurrent validity with other validated measures, and measurement invariance between males and females are further described. Finally, significant differences in the levels of Worry about War are found in relation to several sociodemographic variables, i.e., gender, occupational status, relationship status, and political orientation. Overall, the results of Studies I and II confirm the validity, robustness, and reliability of the War Worry Scale.

## 1. Introduction

In recent years, with the Russian invasion of Ukraine and the dangerous expansion of conflict in the Middle East, war has returned to the center of the news and contemporary political debate, along with its harmful humanitarian, economic, social, and psychological consequences ([11]; [132], [133]). In a postmodernity already characterized by a hyper-acceleration of change ([3]; [2]; [14]; [29]) and collective phenomena of traumatic impact—such as economic crises, pandemics, and climate change—war emerges as a potential cumulative factor of traumaticity ([72]) that undermines the socioeconomic fabric of states, the sense of security, community belonging, and social bonds ([97]; [57]).

On the European continent, after 70 years of peace and prosperity, the return of war occurred while people were still grappling with the economic, health, and psychological effects of the pandemic trauma ([5]; [59]; [40]; [106]; [127]), slowing the recovery process ([99]). With the Russian–Ukrainian war, a new historical period has begun ([118]), marked by humanitarian and economic crises ([2]; [10]; [130]), and the re-emergence of the nuclear power as a threat, fueling anxieties, fears, and worries far beyond the geographical areas directly involved ([112]; [123]). In this scenario of uncertainty, the past year has seen a dangerous escalation in the Middle East, with a tragic legacy of destruction, death, and injury, particularly among civilians ([63]); the reappearance of previously eradicated diseases; and a food crisis that has turned Gaza into a genuine humanitarian emergency ([61]; [134]), with the potential to spread to Lebanon.

The impact of wars has mainly been analyzed in terms of their degree of destructiveness and the economic and territorial transformations they have produced ([97]). The psychological and social consequences, however, are more difficult to capture. Nevertheless, wars are undeniably traumatic events due to their destructiveness, violence, and uncontrollability ([4]). Psychological research has focused on the effects of conflict on directly affected communities, consistently highlighting its role in increasing post-traumatic, depressive, anxiety, and psychosomatic symptoms in different contexts ([1]; [16]; [27]; [28]). The negative impact of war on the mental health of the population, especially children, adolescents, and young adults, has also been highlighted in recent studies conducted in the context of the Russian–Ukrainian war and in the Gaza Strip ([76]; [94]; [112]; [138]; [121]; [126]).

### 1.1. “Beyond the Bombs”: The Indirect Psychological Impact of the War

In 1917, [7] ([7]) emphasized the importance of considering the effects of war even in contexts not directly involved, a point that is reiterated by [128]’s ([128]) theory of multilevel trauma, according to which —beyond the individuals directly experiencing a traumatic event —specific individual characteristics, media coverage of the event, or identification dynamics with victims may play a role in the development of anxiety and post-traumatic symptoms in individuals or communities not directly involved in the trauma (levels 5 and 6). Although research in this area is still limited, psychological studies have increasingly examined the indirect effects of war since the invasion of Ukraine ([117]). These studies, ranging from the cross-cultural study by [34] ([34]) to those conducted in Germany and Poland ([52]; [68]), have highlighted the extent to which the outbreak of conflict has fueled negative emotions and various forms of mental suffering, and how fear of war increases psychological distress, which, as noted above, particularly affects women and the youth population ([54]; [103]).

Among the developmental targets at risk are young adults, who are already deeply affected by the COVID-19 pandemic ([8]; [101]; [106]; [135]) and by pervasive worries about the future of the world ([92]). The pandemic has indeed contributed to exacerbate the psychological distress that characterizes this developmental target group ([98]; [114]), clearly highlighting the “psychic fragility” ([40]) fostered over the years by the dominant cultural logics ([12]; [33]; [66]). The experience of the lockdown, the uncertainty and fear fueled by the uncontrollability of the pandemic’s progression, the reduction in social interactions, and the increase in socioeconomic difficulties in many households have deeply affected the mental health of young adults, leading to increased anxiety, depressive symptoms, stress-related problems, and concerns about their personal and societal future ([80]; [101]; [102]; [106]; [116]; [119]). The increase in these forms of distress has resulted in a true mental health emergency in Italy ([6]) and highlighted how the pandemic experience has contributed to a particular vulnerability among young people to subsequent collective events of a potentially traumatic in nature, such as war and the climate crisis ([9]; [80]; [95]; [110], [111]; [139]).

Consistent with findings from studies conducted in other contexts ([52]; [68]), recent research has shown high levels of fear of war among young Italian adults, which predicts higher levels of self-reported psychological distress ([108]). Furthermore, studies have highlighted the mediating role of intolerance of uncertainty and an anxious view of the future in this relationship ([109]). Furthermore, war is currently highly present in media communication, especially through the dissemination of information, images, and videos that—by brutally depicting the destruction and suffering of populations in conflict—increase uncertainty, anxiety, and worry about war in various contexts ([34]; [49]; [84]; [65]). As previously highlighted in the context of the COVID-19 pandemic ([46]; [88]), the media explosion of distressing events, interacting with various individual characteristics such as a predisposition to anxiety or worry traits or a pessimistic view of the future, can exacerbate psychological distress in young adults ([100]; [137]). For this reason, discussing war in contexts not directly involved in conflicts has become a current necessity, as evidenced by the European Commission (2022) and several recent studies conducted in Italy ([9]; [15]; [95]), which found that the European population is most concerned about the escalation of conflicts, their negative economic consequences, and the possible involvement in a nuclear war.

### 1.2. From the Complex Mental Process of Worrying to the Worry About War

When faced with an event or problem that is perceived as stressful, threatening, and whose outcome is unpredictable, uncertain, and/or negative, the mental process of worry is crucial for understanding and defining the stressor and selecting strategies for coping with it ([22]). It is defined as “a chain of thoughts or images burdened with negative emotions […] that is relatively uncontrollable” ([22]), the result of appraising a stressor that, if perceived as threatening and as a harbinger of consequences—such as war—is likely to foster psychological distress ([79]). Conceptualized as a mental process focused on negative verbal thoughts rather than images ([91]), research has gradually differentiated the construct of worry from anxiety and fear, despite its strong associations with and role in anxiety and depression ([21]; [38]; [48]).

As reflected in the definition, all individuals experience worry as an adaptive process that supports problem solving, focuses attention on the perceived problem/threat, and implements action ([38]). It takes on a disabling function when it becomes an uncontrollable and persistent process that negatively affects everyday life, interferes with problem solving, and promotes avoidance and procrastination strategies ([58]; [124]). Research has shown that not only in its maladaptive function, but also as everyday worries—worries driven by daily and contextual stressors—worry can impair psychological well-being, fueling anxiety, stress, depression, psychosomatic symptoms, dissatisfaction with life, and feelings of loneliness, particularly in the developmental target of young adults ([20]; [48]; [69]; [30]; [71]; [70]; [24]; [131]).

War is a contextual stressor of a macrosocial nature ([18]), the destructiveness and violence of which supports the perception of the event as a threat and fuels various concerns related to its immediate and future consequences ([113]). The worries generated by the war in countries not directly involved ([9]; [15]; [43]) could be related to the undermining of the need for safety, security, and belonging that collective traumatic experiences compromise ([57]).

In line with the aforementioned literature, Worry about War has been considered as a cognitive–emotional process, mainly characterized by verbal–linguistic thoughts (and less by images) about war and its possible effects in the immediate (present) and long-term (future). In line with the dual nature of worry (adaptive/disadaptive), worry about war, on the one hand, could lead to awareness of the phenomenon, information seeking, and the adoption of behaviors to support affected populations, but, on the other hand, it can also become a persistent and/or uncontrollable mental process that, in interaction with complex negative emotions, can lead to psychological distress; it is also a form of apprehension about the consequences that war has or could have on oneself, loved ones, and other peoples.

Although several studies have explored the relationship between wars, increased worry, and mental health ([47]; [95]; [54]), to our knowledge, there are no instruments that specifically assess the worry about war. In general, only a few instruments are currently available to measure the indirect psychological effects of war (Fear of War Scale by [68]; ad. it. [108]; War Anxiety Scale by [123]; War-related Stress Scale by [136]), especially when considering the Italian context. Therefore, as [123] ([123]) argue, precisely because of the relevance of the field of investigation, there is a need for valid and reliable measures to investigate the negative effects of war in different cultural contexts, and to identify the most vulnerable targets. The present study attempts to address this need by developing a scale of the impact of war, as measured by the construct of worry, to fill the gap in the literature. Specifically, worry—due to its association with a more enduring reflective component ([42]; [48])—may be a more suitable construct than anxiety or fear for assessing the impact of collective phenomena such as war in contexts not directly involved in the conflict but still strongly influenced by its media, social, and economic effects.

### 1.3. Phases and Aims of Empirical Research Design

The present study describes the process of developing, validating, and evaluating the psychometric properties of the War Worry Scale (WWS), an instrument designed to assess worry about war in contexts not directly involved in war.

The first phase was devoted to the process of construct definition, item construction, and instrument design, followed by an evaluation of the relevance of the pool items by a panel of experts and a pilot test with a group of young adults to assess the clarity and comprehensibility of the generated items.

The first version of the WWS was then administered to a first sample of young Italian adults in order to examine the item characteristics, the dimensionality of the scale, and some preliminary psychometric properties (Study I). Subsequently, a second sample of young Italian adults was recruited in order to confirm the factorial structure that had emerged in the previous study, to test the internal consistency of the WWS, to explore the measurement invariance of the instrument with respect to gender, and to verify its convergent, discriminant, and concurrent validity (Study II). Figure 1 provides a graphical representation of the steps involved in the development and validation of the WWS.

## 2. The War Worry Scale (WWS)

### 2.1. Construct Definition, Item Generation, and Scale Design

Following the recommendations of [17] ([17]), the construction of the WWS followed the subsequent steps: (a) construct definition; (b) item generation and questionnaire design; (c) questionnaire pilot testing to verify the quality and comprehensibility of the measure; (d) questionnaire administration, latent factor exploration, and item purification; (e) construct validity testing ([17]; [120]).

After defining the construct of Worry about War through a careful examination of the literature described previously, WWS items were constructed to assess the frequency of worry about war in a sample of young Italian adults, integrating inductive and deductive approaches for this purpose ([17]). As part of a larger research project exploring the impact of contemporary collective phenomena on the mental health of young Italian adults, the individual and collective worries of 200 young adults (*M* = 21.10; *SD* = 2.09; 69.0% female; 30.5% male; 0.5% non-binary) were collected in anonymous narrative form in December 2023 and January 2024 via an online form. The form was created to allow for young participants to express their worries in writing, based on two broad narrative prompts: “Tell us about your worries regarding the personal sphere” and “Tell us about your worries regarding the collective sphere (what is happening in the world)”. The collected material was subjected to a content analysis in order to understand participants’ experiences and subjective meanings ([75]). With regard to the area of Collective Worries, i.e., relating to contemporary collective events, the results revealed a central theme named “A Polytraumatic Era”, in which the recurrence and relevance of war led to the creation of the thematic subcategory “Worry about War”. The narrative excerpts associated with this subcategory (see Appendix A) guided the definition of the WWS items.

In an integrated manner, the following were also considered in the construction of the items:

(a) Recent research and reports on the psychological impact of war in non-directly involved contexts, mainly in Europe and Italy ([9]; [15]; [43]; [34]; [52]; [68]; [108]).

(b) Existing instruments on the psychological impact of war in contexts that are not directly involved ([68]; [136]; [108]), as well as instruments on the same worry construct but oriented towards investigating different domains of interest, such as the Climate Change Worry Scale and the Penn State Worry Scale ([122]; [93]).

The Italian adaptation of the Fear of War Scale has been the reference instrument in the construction of the WWS, although the results of previous studies have shown the usefulness of constructing an ad hoc instrument has become apparent. In fact, it is not only the focus on the worry construct that, as noted above, seems more specific to the study of the impact of war on the mental health of young adults not directly involved in war. Results from previous studies using the Fear of War Scale ([108], [109]) have shown an interesting difference in the scores reported on the two dimensions investigated: the Physiological dimension of fear has much lower mean scores than the Experiential dimension of fear. This finding is understandable, of course, given the specificity of the population studied, for whom the physiological effects associated with war (tremors, heart palpitations, insomnia, etc.) appear to be much less relevant, as these are subjects not directly involved in war contexts. In order to focus respondents’ attention on the construct of interest and to avoid confusion with terms that are often used interchangeably in common language (e.g., anxiety and fear), and following the main specific instruments for measuring worry ([93]), all constructed items contained expressions such as “I worry”, “I am worried”, and “A worry of mine”. At the same time, in order to focus attention on the domain of interest, the word “war” was included in all items of the instrument except for one, where the term “wars” was replaced by the synonym “conflicts”.

The scale is designed to be a short and flexible instrument that measures self-reported levels of worry about war while encouraging reflection on the topic. The content of the items was constructed to capture worry about war, a construct designed to integrate both worry about the present and the tangible effects of conflict (e.g., the consequences of war in affected countries, the tendency to worry) and worry about possible future effects (e.g., consequences for one’s own and others’ future, escalation of conflict, outbreak of nuclear war).

An initial pool of 20 items was constructed with a 5-point Likert response mode (1 = not at all; 2 = a little; 3 = to some extent; 4 = a lot; 5 = very much).

### 2.2. Content and Face Validity

The content validity of the item pool was tested to understand how well it reflected the construct. Three independent experts (one expert in quantitative analysis and two clinical psychologists) were selected to rate the relevance of each item using a 3-point scale (1 = essential; 2 = useful but not essential; 3 = not necessary) ([104]). Sixteen items met the threshold of representativeness among the experts and were considered necessary; four items were defined as unnecessary and eliminated because their content—related to the intrusiveness of the worry and its impact on the future, such as the economic and political consequences of the war—was considered redundant. On the advice of the experts, the item “I am worried about the outbreak of World War III or a nuclear war” was split into two separate items to ensure greater consistency of content.

The 17-item pool obtained after expert review was then preliminarily administered to 13 young Italian adults aged 18 to 30 (*M* = 25.67; *SD* = 3.03; 6 females and 7 males). The purpose of this phase—aimed at testing the face validity of the WWS—was to explore how clear and understandable the constructed items and the whole instrument were to a target population similar to the later recruited samples. Participants were asked to provide feedback (yes or no) on the clarity and comprehensibility of the presentation; the simplicity of the syntax and vocabulary used in the items; the presence of any grammatical errors, inaccuracies, and misunderstandings; the use of the response range; and the overall completion of the scale. Each participant was given the opportunity to report any misunderstandings and/or suggestions for corrections/changes in the space provided. From this pilot test, minor changes were made to the items and administration to promote greater clarity and readability.

## 3. Study I

### 3.1. Materials and Methods

#### 3.1.1. Sample Size Determination

The ratio of subjects per item was used to plan a priori the minimum number of participants based on the data analysis to be performed. The criterion of 10 subjects per item was chosen ([96]; [129]). Thus, a minimum sample size of 170 participants was adequate to evaluate the preliminary latent structure of the scale.

#### 3.1.2. Participants and Procedure

For the Study I, a sample of 250 young Italian adults (40.4% male; 59.6% female) aged between 18 and 30 years (*M* = 22.58; *SD* = 3.04) was recruited. At the time of data collection, most participants lived in Southern Italy (91.6%), specifically in the Campania (82.0%) region of Southern Italy; 59.2% lived in cities and 40.8% lived in the countryside. In terms of relationship status, 51.2% of the participants were single and 48.8% were in a relationship. Most of the participants were students (62.0%), 17.2% were working students, 15.6% were employed, and 5.2% were unemployed. In terms of the educational level, 72.4% of participants had a high school diploma, 16.4% had a bachelor’s degree, 9.2% had a master’s degree, and 2.0% had primary school diploma. Most of the participants had a left-wing political orientation (49.2%), 36.8% were not interested in politics, 10.0% had a centrist political orientation, and 4.0% had a right-wing political orientation.

Participants completed the WWS online using an internet-based survey between May and June 2024. All participants, recruited using convenience and snowball sampling, who were interested in this study were asked to identify other potential respondents in their social network. All participants were adequately informed about the aims of this study, the anonymity of the data collected, and the voluntary nature of participation. All study participants signed the informed consent form on the first page of the survey and were encouraged to answer as truthfully as possible.

#### 3.1.3. Data Analysis

Descriptive statistical analyses were preliminarily implemented to examine the means and standard deviations of each item. The data distribution was explored by assessing skewness and kurtosis. Values between −1.5 and +1.5 were considered indicative of a normal distribution of the data ([125]). The assumption of multivariate normality was also verified using Mardia’s skewness and kurtosis tests ([85]). The results of these preliminary analyses guided the subsequent correlational analyses and the selection of the factorial extraction criterion. Measure of sampling adequacy (MSA ≥ 0.50) was also evaluated to explore whether the pool items measured the same domain ([81]).

Exploratory factor analysis (EFA) was used to explore the latent structure of the WWS. In EFA, the factors are extracted without specifying the number and pattern of loadings between the observed variables and the latent factor variables ([19]). Prior to conducting the EFA, the suitability of the data for the EFA was assessed. Kaiser–Meyer–Olkin (KMO) and Bartlett’s test of sphericity (*χ*^2^) were used. A KMO value between 0.80 to 1 indicates that the sampling is adequate ([67]; [125]). The significant value of the Bartlett’s test of sphericity (*p* < 0.001) indicates that a factor analysis may be worthwhile for the data set ([50]).

EFA was conducted on the 17 items with the robust maximum likelihood (MLM) estimator and the goemin rotated solution. The MLM is a robust variant of maximum likelihood ([13]) and provides robust standard errors. The MLM is referred to as the Satorra–Bentler chi-squared test (SB*χ*^2^) and is used to assess model fit. The comparative fit index (CFI), the Tucker–Lewis index (TLI), the root mean square error of approximation (RMSEA), the standardized root mean square residual (SRMR), and the ratio of SB*χ*^2^ to degrees of freedom (*df*) were also used to evaluate model fit. Goodness of fit was assessed using the following criteria: CFI and TLI, with values between 0.90 and 0.95 were considered good ([25]; [74]), values above 0.95 were considered excellent ([60]), RMSEA and SRMR of approximately 0.80 or less ([26]) were considered acceptable, and a *χ*^2^/*df* ratio value of 3 or less was considered good. Two steps were taken to examine the items and select the best factorial solution. In the first step, the factor loadings of each item on each factor resulting from the EFA were examined. Items that did not load significantly on any factor (<0.50; [37]), items with cross-loadings (≥0.40), and items that showed a difference of less than 0.20 between the primary and alternative factor were removed ([56]). Furthermore, the communalities above 0.40 for each item are acceptable ([37]). In the second step, following the principles of parsimony in scale development, which suggest that a quality factor is composed of four to six items ([56]), the items with the highest factor loading on the extracted factors were selected ([89]).

Finally, Cronbach alpha (α) and McDonald omega (ω) coefficients were carried out to test the reliability of the instrument.

All statistical analyses were performed using SPSS software v. 29 ([62]) and RStudio v. 4.4.1 ([115]) with MplusAutomation package v. 1.1.1 ([53]).

### 3.2. Results

#### 3.2.1. Preliminary Descriptive Statistics of WWS Items

As shown in Table 1, the descriptive skewness and kurtosis indices for each item fall within the range of the normal distribution. At the same time, Mardia’s test was not significant for skewness (*p* > 0.05) but significant for kurtosis (*p* < 0.05), indicating a slight deviation from the normality of the distribution. The MSE indices were above the threshold value of 0.50, indicating that all items were suitable for the latent construct detection. Table 1 shows the means, standard deviations, skewness, kurtosis, variance, item-total correlation, and MSE value for all items of the WWS.

#### 3.2.2. Latent Structure of the WWS: Exploratory Factor Analysis

As for the preliminary tests, the KMO value was 0.93 (95%; CI: 0.89, 0.93), confirming the suitability of the study sample for EFA. The Bartlett’s test of sphericity was significant [*χ*^2^ (136) = 2505.01 (*p* < 0.001)], indicating that the correlation matrix was fully adequate for EFA. The analyses suggest a two-factor solution with the following model fit: SB*χ*^2^ (103) = 301.550; SB*χ*^2^/*df* = 2.928; CFI = 0.899; TLI = 0.866; RMSEA = 0.088 (90% CI [0.076–0.099]); SRMR = 0.045.

The item inspection first looked for items with inadequate saturation and cross-loading. In this step, items 7 and 13 were removed for low saturation on both factors, and items 1 and 12 were removed for cross-loading. Finally, to achieve the optimal solution, items 3, 15, and 16 were removed, leaving the items with the best saturation on the two dimensions and then five items for each dimension.

The first factor, consisting of five items, was labelled “Worry about the present” (WWP) and the second, consisting of five items, was labelled “Worry about the future” (WWF).

Table 2 synthesized the EFA results of the retained items of the factor. The fit of the model is satisfactory with this factor solution: SB*χ*^2^ (26) = 47.147; SB*χ*^2^/*df* = 1.813; CFI = 0.984; TLI = 0.972; RMSEA = 0.057 (90% CI [0.030–0.083]); SRMR = 0.023. The cumulative variance explained was 0.57.

#### 3.2.3. Reliability of the WWS

The Cronbach alpha (α) and McDonald omega (ω) coefficients for the 10-item instrument were 0.902 and 0.898, respectively. These values indicate a good internal consistency of the WWS.

## 4. Study II

### 4.1. Materials and Methods

#### 4.1.1. Sample Size Determination

As in Study I, the ratio of 10 subjects per item was used to plan a priori the minimum number of participants based on the data analysis to be performed ([96]; [129]). In addition, the recommendation that 500 observations can be considered very good for conducting confirmatory factor analysis was taken into account ([83]).

#### 4.1.2. Participants and Procedure

A sample of 500 young Italian adults (39.4% male; 60.6% female) aged between 18 and 30 years *(M* = 22.84; *SD* = 3.04) was recruited for the Study II. Most of the participants lived in southern of Italy (85.0%), especially in Campania (77.0%). The sociodemographic characteristics of this sample are shown in Table 3.

Participants in this study were recruited in Italy via social media sites, through self-report questionnaires using an internet-based survey. Data collection for Study II took place from July to September 2024. The questionnaire was advertised with posters in social spaces at the University of Naples and, as in the previous study, a snowball sampling method was used and all participants interested in the research were asked to identify other potential respondents in their social network. All participants were informed about the aims of this study and the rules regarding privacy and anonymity of the data collected. Participation was voluntary and each participant was free to leave at any time. The inclusion criteria for the present study were as follows: being between 18 and 30 years old, being a resident of Italy, and providing consent to participate in this study on the first page of the online questionnaire.

#### 4.1.3. Instruments

A *sociodemographic section* to examine information on the participants’ age, gender, residence and type of residence, civil status, educational level, occupational status, political orientation, having faith in a God, and participation in peace associations (see Table 3).

The *War Worry Scale (WWS)*, in the form obtained from the analysis of Study I, was used to assess worry about war, consisting of 10 items with a 5-point Likert scale ranging from 1 (not at all) to 5 (very much). This instrument is a self-report instrument designed to measure the Worry about War along the following dimensions: Worry about the Present (WWP) and Worry about the Future (WWF). The first dimension assesses the worry about war through items that focus on current worry about war, its intensity, and the effects of wars on those directly affected; the second dimension assesses worry about war along through items that explore the consequences of wars on one’s future and loved ones, and the possible escalation of wars into wider and/or nuclear conflicts. The numbering and reordering of the items across the two dimensions was corrected for the validation process by CFA completion of the WWS with 10 items.

The *Fear of War Scale* (FOWARS; [68]; Ad it. [108]) was used to assess the fear of war through 12 items with a 5-point Likert scale ranging from 1 (strongly disagree) to 5 (strongly agree). This self-report instrument assesses the Fear of War along two dimensions: the Experiential dimension of fear (item example: “I am afraid the world will no longer be a safe place”) and Physiological dimension of fear (item example: “I start to tremble when I think the war reaches here, as well”). The scale also provides a total score, with a score above 2.5 indicating that the participant is very likely to experience fear of war. The authors of the Italian adaptation reported good psychometric properties and internal consistency ([108]). In this study, Cronbach’s α and McDonald’s ω were, respectively, 0.80 and 0.80 for the Experiential dimension, 0.92 and 0.92 for the Physiological dimension, and 0.89 and 0.89 for the global scale.

The *Dark Future Scale* (DFS; [64]) was used to assess future anxiety through five items that explore concern and anxiety about the future, considering the cognitive and emotional processes that induce fear for the future to dominate over hope. This measure is a 7-point Likert scale ranging from 0 (=definitely untrue) to 6 (=definitely true). The total score ranges from 0 to 30, with higher scores indicating higher levels of Future Anxiety. The scale revealed excellent psychometric properties ([64]), and in this study, Cronbach’s α and McDonald’s ω were 0.88 and 0.88, respectively.

The *Resilience Scale* (CD-RISK-10; [36]) was used to measure resilience through 10 items that assess hardiness, flexibility, emotion regulation ability, cognitive focus under stress, and sense of self. The scale has a 5-point Likert response scale ranging from 0 (=not true at all) to 4 (=true nearly all the time). The total score ranges from 0 to 40, with higher scores indicating greater resilience. The CD-RISK-10 reported good internal consistency ([36]) and in the current study’s Cronbach’s α and McDonald’s ω were 0.89 and 0.89, respectively.

The *Depression, Anxiety, and Stress Scale* (DASS-21; [82]; [23]) was used to assess psychological distress. Across 21 items with a 4-point Likert-type scale ranging from 0 (did not apply to me at all) to 3 (applied to me very much, or most of the time), the scale assesses depression, anxiety, and stress in the last 7 days by using three subscales composed of stress (item example: “I felt like I had nothing to look forward to”), anxiety (item example: “I felt close to a panic attack”), and depression (item example: “I found it hard to relax”). The scale also provides a total score as an index of self-reported global psychological distress. The authors of the Italian version of the DASS-21 reported good internal consistency ([23]). In the present study, Cronbach’s α and McDonald’s ω were 0.90 and 0.90 for Stress, 0.88 and 0.88 for Anxiety, 0.90 and 0.91 for Depression, and 0.95 and 0.95 for the global score, respectively.

#### 4.1.4. Data Analyses

Descriptive statistical analyses were first carried out to examine the means and standard deviations of each item. The distribution of the data was examined by assessing skewness and kurtosis. Values between −1.5 and +1.5 were considered indicative of a normal distribution of the data ([125]). The assumption of multivariate normality was also tested using Mardia’s skewness and kurtosis tests ([85]). To confirm the factorial structure, confirmatory factor analysis (CFA) was used. In line with the EFA solution (see Study I), a two-factor solution was specified. Moreover, also an overarching factor (second-order factorial structure) was specified. The hierarchical model implies that each item loaded onto its specific first-order factor which, in turn, loaded onto an overarching general factor.

The MLM estimator was used. Model fit was assessed by considering the aforementioned fit indices: CFI, TLI, RMSEA, SRMR, and the ratio of SB*χ*^2^ to degrees of freedom. The following criteria were used: CFI and TLI, with values between 0.90 and 0.95 were considered good ([25]; [74]), values above 0.95 were considered excellent ([60]), RMSEA and SRMR approximately 0.80 or less were considered acceptable ([26]) and a *χ*^2^/*df* ratio value of 3 or less was considered good.

Measurement invariance (MI) analyses were computed to evaluate whether the factorial structure of the WWS was invariant between females and males. Three nested models were run sequentially, constraining the parameters the model to be equal between males and females. First, the factor structure was constrained to be equal in the two groups (Model 1: Configural Invariance). Second, the factor loadings were constrained to be equal in the two groups (Model 2: Metric Invariance). Third, factor loadings and intercepts were constrained to be equal in the two groups (Model 3: Scalar Invariance). The model fit of each model was evaluated using the aforementioned indices and cut-offs: CFI and TLI, with values between 0.90 and 0.95 were considered good ([25]; [74]) and values above 0.95 were considered excellent ([60]); RMSEA and SRMR approximately 0.80 or less were considered acceptable ([26]), and a *χ*^2^/*df* ratio value of 3 or less was considered good.

The assumption of invariance was assessed using the SB*χ*^2^ DIFFTEST, the |ΔCFI|, and the |∆RMSEA|. As the *χ*^2^ statistic may be influenced more by the sample size of the comparison groups and less by a lack of invariance, the |ΔCFI| has been recommended as the best approach to assess measurement invariance ([32]). Therefore, |∆CFI| (≤0.01) and |∆RMSEA| (≤0.015) were selected along with DIFFTEST. As recommended by [31] ([31]), a worse factorial structure is obtained when two out of three cut-offs overpass.

Then, to test the proportion of variance explained by each latent construct in its corresponding observed variables, the average variance extracted (AVE) was calculated. The AVE value should be greater than 0.50 ([51]). In order to distinguish the ability of the latent variable to discriminate itself from others within the model, in this case Worry about the Present and Worry about the Future, the AVE should exceed the shared variance between constructs, as assessed by the squared factor correlation, according to the criterion of [45] ([45]).

Reliability and validity were also assessed. Reliability was evaluated using Cronbach’s alpha (α) and McDonald’s omega (ω) coefficients. When assessing validity, some studies have suggested the importance of assessing multiple aspects of validity, such as criterion validity and construct validity ([41]), while other authors have suggested that it is also appropriate to consider validity as a unified concept, although not a simple one ([73]). In this study, validity was assessed by considering convergent, divergent, and concurrent validity. Convergent and divergent validity (construct validity) were assessed using Person’s correlations. Convergent validity indicates whether the instrument is related to others constructs that have been assessed at the same measurement point and that are theorized to be related to worry about war on the basis of theoretical assumptions. Divergent validity indicates whether the instrument is not correlated with other constructs that have been assessed at the same point in time and that are not theorized to be related to worry about war on the basis of theoretical assumptions. Concurrent validity (criterion validity) involves the selection of a criterion to indicate an intended or subsequent outcome.

Finally, *T*-test for gender differences and ANOVA with Tukey post-hoc tests analyses for sociodemographic differences (occupational status, relationship status, political orientation, and type of course of study) for the general dimension of Worry about War. Effect sizes were measured through Cohen’s *d* (small ≤ 0.02; medium = 0.05; large ≥ 0.08; huge ≥ 1.0) and Eta-square (*η*^2^; small ≥ 0.01; medium ≥ 0.059; large ≥ 0.138) ([35]).

All statistical analyses were performed using SPSS software v. 29 ([62]) and RStudio v. 4.4.1 ([115]) with MplusAutomation package v. 1.1.1 ([53]).

### 4.2. Results

#### 4.2.1. Confirmatory Factor Analysis

Means, standard deviations, skewness, and kurtosis values for all WWS items are shown in Table 4. The results indicate an adequate distribution of the data. The second order model showed an excellent fit of the data (Figure 2). Specifically, although the chi-square statistic was statistically significant, the fit indices indicated an excellent fit to the data [SB*χ*^2^ (34) = 92.601; *p* < 0.001; SB*χ*^2^/*df* = 2.723; CFI = 0.973; TLI = 0.965; RMSEA = 0.059, (90% C.I. [0.045–0.073]); SRMR = 0.035.

All factor loadings were statistically significant and loaded on the specific first-order factor. All items ranged from 0.671 (Item 9) to 0.844 (Item 1) (Table 4). Correlations among latent factors (first order, WWP and WWF) were positive and significant *(r* = 0.682).

#### 4.2.2. Measurement Invariance

In a first step, the second-order model was specified separately for females and males. For the female model, goodness of fit indices revealed an excellent fit to the data: SB*χ*^2^ (34) = 77.581; *p* < 0.001; SB*χ*^2^/*df* = 2.28; CFI = 0.967; TLI = 0.956; RMSEA = 0.065, [0.046–0.080] 90% C.I.; SRMR = 0.042. For the male model, the goodness of fit indices revealed an excellent fit of the data: SB*χ*^2^ (34) = 54.461; *p* < 0.001; SB*χ*^2^/*df* = 1.602; CFI = 0.975; TLI = 0.968; RMSEA = 0.055, 90% C.I. [0.025–0.080]; SRMR = 0.038.

In a second step, using the total sample, the measurement invariance was tested (Table 5). Configural invariance was specified. The model showed excellent goodness of fit indices—SB*χ*^2^ (68) = 131.455; *p* < 0.001; SB*χ*^2^/*df* = 1.933; CFI = 0.971; TLI = 0.961; RMSEA = 0.060, 90% C.I. [0.045–0.077]; SRMR = 0.041—suggesting that the factor structure was equal between females and males. Then, metric invariance was specified. The model showed an excellent fit of the data: SB*χ*^2^ (76) = 143.548; *p* < 0.001; SB*χ*^2^/*df* = 1.889; CFI = 0.969; TLI = 0.963; RMSEA = 0.060, 90% C.I. [0.044–0.074]; SRMR = 0.048. A non-statistically significant decrease in DIFFTEST (=11.085; *df* = 8; *p* = 0.197) and a non-statistically significant decrease in |ΔCFI| = 0.002 and |ΔRMSEA| = 0.000 were found, indicating that the items were equivalently related to the latent factor independently of gender. Then, scalar invariance was specified. The model showed excellent goodness of fit indices: SB*χ*^2^ (84) = 169.545; *p* < 0.001; SB*χ*^2^/*df* = 2.018; CFI = 0.960; TLI = 0.957; RMSEA = 0.064, 90% C.I. [0.050–0.078]; SRMR = 0.053. Despite the difference in the DIFFTEST (=24.995; *p* < 0.001), a non-statistically significant decrease in |ΔCFI| = 0.009 and |ΔRMSEA| = 0.004 was found, indicating that the same expected item response at the same absolute level of the trait was obtained in both female and male samples. Table 5 details the model comparison.

#### 4.2.3. Reliability and Validity

In terms of internal consistency, Cronbach’s alpha and McDonald’s omega indices showed good internal consistency for each scale, WWP and WWF, and for the total score. For the WWP, Cronbach’s alpha and McDonald’s omega were 0.875 and 0.875, respectively. For the WWF, Cronbach’s alpha and McDonald’s omega were 0.858 and 0.858, respectively. For the WW (general factor), Cronbach’s alpha and McDonald’s omega were 0.897 and 0.893, respectively.

As shown in Table 6, the AVE for each scale, WWP and WWF, exceeds the cut-off value of 0.50. The squared correlations between the WWP and WWF were lower than their respective AVE values, supporting the absence of a multicollinearity problem in the measure and confirming internal validity.

The correlations presented in Table 7 show the convergent, divergent, and concurrent validity of the instrument. Specifically, the results show a significant and positive relationship between the WW dimensions and the dimensions of Fear of war, WWP, and Experiential dimension of fear (*r* = 0.66); WWP and Physiological dimension of fear (*r* = 0.37); WWF and Experiential dimension of fear (*r* = 0.65); and WWP and Physiological dimension of fear (*r* = 0.55), confirming the convergent validity of the instrument. The lack of association between the WW dimensions and resilience (*r* = 0.04 and *r* = 0.03, respectively) confirmed the divergent validity of the instrument. Finally, the relationship between the WW dimensions and general distress (*r* = 0.23 and *r* = 0.21, respectively) and the WW dimensions and Future Anxiety (*r* = 0.31 and *r* = 0.23, respectively) confirmed the concurrent validity of the instrument.

#### 4.2.4. Group Differences

Analyses for sociodemographic differences in worry about war revealed differences between females and males, occupational status, and type of study course. Specifically, a *t*-test showed significant gender differences, with higher scores in Worry about War in female than males (*M*_Female_ = 3.65 vs. *M*_Male_ = 3.177; *t*_(498)_ = 6.45; *p* < 0.001; *Cohen’d* = 0.59). ANOVA with Tukey post-hoc test showed a significant difference regarding occupational status (*F*_(3, 496)_ = 3.26; *p* < 0.05; *Cohen’d* = 0.02), with students experiencing greater Worry about War than workers *M*_Students_ = 3.52 vs. *M*_Workers_ = 3.21. Regarding to the students, *t*-test analysis showed that students of humanistic courses had significantly higher levels of Worry about War than students of scientific courses (*M*_HStudents_ = 3.60 vs. *M*_SStudents_ = 3.33; *t*_(383)_ = 3.25; *p* < 0.001; *Cohen’d* = 0.34). *T*-test analyses also showed significant differences by relationship status. Specifically, young adults engaged in a romantic relationship reported significantly higher levels of Worry about War than those who were singles (*M*_Relarions_ = 3.53 vs. *M*_Single_ = 3.39; *t*_(448)_ = 1.90; *p* < 0.05; *Cohen’d* = 0.17). Similarly, young adults involved in peace associations showed higher levels of worry about war than those not involved (*M*_PeaceAss.Yes_ = 3.81 vs. *M*_PeaceAss.No_ = 3.43; *t*_(498)_ = 2.58; *p* < 0.01; *Cohen’d* = 0.45). Finally, ANOVA analyses showed significant differences in political orientation (*F*_(3, 499)_ = 6.93; *p* < 0.001; *Cohen’d* = 0.04), with young people who described themselves as left-wing having higher levels of worry about war than those who described themselves as centrist (*M*_Left_ = 3.61 vs. *M*_Centre_ = 3.28; *p* < 0.05) and those who described themselves as not interested in politics (*M*_Left_ = 3.61 vs. *M*_NotInterested_ = 3.29; *p* < 0.001).

## 5. Discussion

The present study describes the process of development and validation of the War Worry Scale (WWS), an instrument designed to assess the worry about war in non-war-torn environments, such as Italy. The development of the WWS aims to fill the gap in the literature regarding the lack of instruments to assess the indirect psychological effects of war. For this purpose, the construct of worry was chosen, which is defined as a chain of thoughts or images characterized by negative emotions ([22]), resulting from the evaluation of a perceived stressor as threatening and heralding consequences ([79]). In line with this literature and with [128]’s ([128]) theory, worry about war has been defined taking into account that this phenomenon can be understood as an event that can have multilevel traumatic effects and can also affect the mental health of individuals and communities indirectly exposed to it. In order to study the effects of war in this type of context, the construction of appropriate and valid measures is seen as a need ([123]), to which this study aims to contribute. The final version of the WWS is presented in Appendix B.

The items were generated from a content analysis ([75]) carried out on unpublished narrative material collected with a large group of young Italian adults (inductive approach). The selected narrative excerpts—in continuity with what has been highlighted in the literature ([9]; [15]; [95])—highlighted the involvement of young adults in contemporary issues, and became fundamental for defining the content of the WWS items. At the same time, as recommended by [17] ([17]), the item development and scale design took into account the literature review on the topics of interest and the comparison with other validated instruments related to the worry constructs or the specific domain (psychological impact of war). Expert evaluation and a pilot test with a small group of young Italian adults supported the assessment of the content and face validity of the instrument. Study I explored the latent structure of the WWS by integrating the exploratory factor analysis (EFA). The EFA suggested a two-dimensional structure, highlighting that all items were significantly consistent in defining the two factors, with factor loadings and communalities above the cut-offs considered ([44]). Items 1, 3, 7, 12, 13, 15, and 16 were removed because they did not meet the selected criteria.

The factorial structure emerging from the EFA was confirmed by the confirmatory factor analysis (CFA) conducted on an independent sample of young Italian adults in Study II. The results showed that the WWS can be considered a solid and robust measure in terms of statistical fit, reliability, and validity. CFA results showed a very good fit of the second-order structure with fit indices in line with recommendations in the literature ([60]; [90]). All items loaded in the hypothesized dimension were in line with the results of Study I.

Specifically, the first dimension, Worry about the Present, consists of five items that assess worry about war through items that focus on current worries about war and its intensity (e.g., “Worry about war very often occupies my thoughts”) and on the effects of war on those directly affected (e.g., “I worry when I think about the effects of the war on people living in places of conflict”). The second dimension, Worry about the Future, assesses worry about war through items that explore the consequences of wars for one’s future and loved ones (e.g., “I am worried that the war will compromise my future”) and the possible escalation of wars into wider and/or nuclear conflicts (e.g., “I am worried about the outbreak of a nuclear war”).

Measurement invariance (configural, metrical, and scalar invariance) across gender was confirmed. As recommended by [32] ([32]) and [105] ([105]) when comparing configural, metrical, and scalar invariance models, the alternative indices (ΔCFI and ΔRMSEA) were more considered, as the chi-squared statistic is often affected by the sample sizes of the groups, which in our case were also not homogeneous. The results show that the WWS is an instrument that can measure the same construct for Italian young adults of different genders (male and female). The invariance of the instrument allows for the comparison of the means of the two groups and opens the possibility for future comparative analyses, such as multi-group SEM analysis.

In terms of internal consistency, the results suggest that the WWS is a reliable instrument and that all items contribute significantly to the excellent internal consistency of the scale. Considering [45]’s ([45]) criteria, the instrument showed good internal validity and no multicollinearity between WWP and WWF.

Correlation analyses between the WWS dimensions and the Fear of War Scale support the convergent validity of the instrument. In particular, the positive correlation with the FOWARS, and more specifically with the Experiential dimension of fear, which is specifically referred to in the definition of the WWS items, highlights the relationship between the constructs of worry about war and the fear of war, in line with the literature ([21]; [38]; [48]).

Correlation analyses between WW dimensions and resilience supported the divergent validity. In particular, the lack of correlation with resilience—a construct traditionally understood as a dispositional trait—highlights the good discriminant validity of WWS, which in contrast detects a specific worry driven by a macrosocial and contextual stressor perceived as threatening ([18]; [79]). Finally, correlation analyses between the WWS dimensions and the Dark Future Scale and the Depression, Anxiety, and Stress Scale supported the concurrent validity. This result confirms the findings highlighted in the literature, which, despite their differences, highlight the strong link that worry plays in anxiety and depressive symptomatology ([38]; [48]). At the same time, the results indicate that this positive and significant correlation is not particularly large. This finding is also consistent with the research on worry about climate change ([122]) and highlights the intrinsic nature of the construct of worry, which, in addition to its ability to affect psychological well-being ([58]), may also have an adaptive function in coping with stressors ([38]). Furthermore, recent research suggests that the relationship between emotions and mental processes related to global collective phenomena and mental health outcomes should be explored in greater depth, taking into account the possible influence of additional variables ([122]). Finally, with regard to the relationship between WW and DFS, our findings are in line with recent studies that have highlighted the extent to which collective events fuel worries about the future ([92]) and a negative view of the future ([109], [110]).

Study II also analyzed differences in levels of WWS according to sociodemographic variables. Specifically, the analyses show that females are more worried about the war than males. This finding is in line with several Italian and European studies on the psychological impact of war ([52]; [68]; [109]). Higher levels of concern about the war are also found among young adults attending university, particularly in the humanities, than among people of the same age who are working. On the one hand, this finding suggests that young students are in the process of constructing their future and collective events—such as war, climate change, COVID-19—represent unresolved challenges to be faced ([55]; [78]; [87]); on the other hand, the university context could be a stimulus for learning about the impact of contemporary collective phenomena and this could fuel worry about war and worry about the future. These findings seem to be in line with what is clearly visible in the narrative excerpts collected in a preliminary sample of 200 young adults, reported in Appendix A, where a worry about contemporary traumatic events and their possible impact on the future emerges.

Furthermore, the highest levels of worry about the war were found in couples compared to single participants. This finding could be interpreted by considering that war—like other potentially traumatic contemporary phenomena ([86])—may influence couples’ future planning, particularly the decision to have children in a world perceived as “dangerous”.

The results also show that people who participate in peace organizations and have a left-wing political orientation have a higher level of concern about war. This finding seems to indicate that young adults who participate in peace activities, in line with the mission of the associations, have higher levels of worry about the impact of war in the present and future. Finally, our results suggest that political ideology may also play a role in worry about war. Despite these findings, future studies could deepen these findings by considering numerically more homogeneous groups and provide evidence to guide the construction of interventions to support the target group of young adults.

### Limitations and Future Directions

Some limitations must be acknowledged. Convenience sampling and self-report instruments were used, and known biases—related to individual characteristics and social desirability—may have influenced responses. To overcome these limitations, future studies could use representative samples. In addition, most of the participants were young adult students from Southern Italy. Although the research highlights how this age group is more invested in contemporary issues than others—which is why this target group was chosen—future studies could include a more diverse sample of young adults, involve more young workers, and consider different levels of education in more detail. The factorial structure and the invariance of the instrument were tested by considering young adults mainly from Southern Italy. Future studies could use the instrument with adolescents and the general population. This could be useful in order to make comparisons in Worry about War levels both in different age groups and in different areas of Italy. Furthermore, this study did not consider media exposure to war information as a sociodemographic variable. Exposure may indeed have an effect on war anxiety in young adults in non-war-torn environments. Future studies will address this limitation by considering media exposure as a variable of interest. Finally, other contextual variables could be considered because of their potential impact on worries about war, such as the role of the parents, which has already proven to be significant in the experience of COVID-19 ([106], [107]). On the one hand, parents could provide a safe space for discourse to mitigate anxieties about the future, but on the other hand, they could also fuel these concerns. For example, studies show how helicopter parenting, characterized by overprotection and control, can increase emerging adults’ anxiety and intolerance of uncertainty ([77]), both of which are relevant to war-related concerns. Future studies could take this into account when assessing possible predictors of worry about war.

## 6. Conclusions

The present study increases the availability of a valid and reliable measure to detect the dimension of worry about war in contexts not directly involved in war. The different steps presented in this paper (Figure 1) describe the process of developing and validating the War Worry Scale (WWS), a new instrument for exploring worry about war. The item construction process followed a rigorous integration of inductive and deductive approaches, and subsequent studies revealed the second-order structure of the WWS, factorial robustness, good internal consistency, invariance of the instrument between male and female, and good convergent, divergent, and concurrent validity. The good psychometric properties support the use of the WWS to assess worry about war in young Italian adults, responding to the growing need for valid and reliable psychometric instruments to measure the impact of war in contexts not directly involved in it ([123]). The WWS could be used to explore how contemporary wars affect the mental health of the Italian young adult population, taking into account different risk and protective factors that may play a role in this relationship. Given the growing involvement of young adults in contemporary issues, as highlighted in the literature ([15]; [92]), the WWS could be a valuable tool to guide the design and implementation of targeted interventions to support the target population. Specifically, we believe that the WWS could be useful in individual clinical settings to assess war-related concerns, where appropriate, and thereby tailor psychological interventions to each individual’s unique needs. At the same time, we believe that this tool could also be a valuable aid in group interventions, where young people, under the guidance of an expert, can express, share, and discuss emotions, thoughts, and moods heightened by contemporary conflicts. These groups could provide a safe space for emotional expression while simultaneously supporting adaptive coping strategies, the maintenance of hope, and both individual and group empowerment ([39]). In this regard, the WWS could be integrated into intervention programs aimed at raising awareness of the mental health issues that collective events such as war can generate, becoming a tool capable of stimulating reflection and discussion on these issues, and facilitating the identification of young adults most at risk of developing psychological distress. In this sense, the WWS could be a useful tool for the design of educational and psychological interventions and for the longitudinal evaluation of their effectiveness.

## Figures and Tables

**Figure 1 ejihpe-15-00024-f001:**
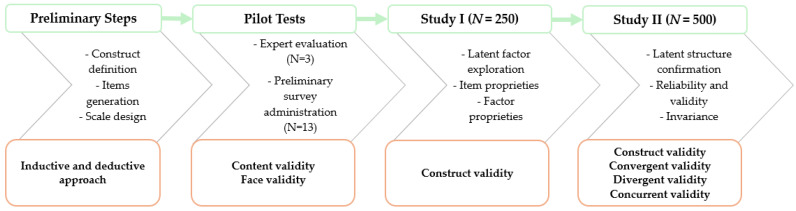
Steps and phases of WWS development and validation.

**Figure 2 ejihpe-15-00024-f002:**
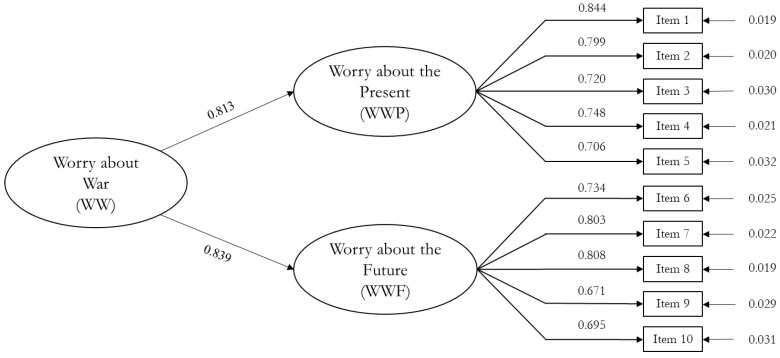
Graphical representation of the WWS model.

**Table 1 ejihpe-15-00024-t001:** Items descriptive Statistics (*N* = 250).

Items	Mean	SD	Skewness	Kurtosis	Variance	Item-Total *r*	MSE
Item 1	3.68	1.10	−0.48	−0.58	1.22	0.69	0.96
Item 2	3.56	1.14	−0.32	−0.81	1.29	0.69	0.92
Item 3	3.80	1.03	−0.69	−0.01	1.05	0.68	0.93
Item 4	3.89	1.05	−0.77	−0.03	1.11	0.76	0.93
Item 5	3.70	1.08	−0.58	−0.39	1.16	0.71	0.95
Item 6	3.06	1.20	0.12	−0.96	1.43	0.61	0.91
Item 7	3.72	1.16	−0.57	−0.61	1.34	0.71	0.95
Item 8	3.16	1.25	0.00	−1.03	1.55	0.66	0.92
Item 9	4.13	1.06	−1.23	0.86	1.13	0.68	0.94
Item 10	2.80	1.14	0.16	−0.73	1.31	0.63	0.92
Item 11	3.79	1.16	−0.72	−0.36	1.35	0.68	0.95
Item 12	3.96	1.11	−0.97	0.13	1.23	0.70	0.90
Item 13	2.83	1.22	0.23	−0.89	1.48	0.41	0.90
Item 14	3.23	1.22	−0.11	−1.01	1.49	0.64	0.92
Item 15	3.56	1.18	−0.53	−0.55	1.40	0.66	0.94
Item 16	3.92	1.15	−0.88	−0.10	1.33	0.51	0.94
Item 17	3.65	1.26	−0.54	−0.83	1.58	0.64	0.88

**Table 2 ejihpe-15-00024-t002:** Item factor loadings and communalities.

Items	Factor 1	Factor 2	Communalities
Item 2		0.535	0.465
Item 4	0.872		0.767
Item 5	0.772		0.643
Item 6		0.771	0.600
Item 8		0.892	0.724
Item 9	0.761		0.550
Item 10	0.793		0.582
Item 11	0.723		0.529
Item 14		0.554	0.421
Item 17		0.770	0.597

**Table 3 ejihpe-15-00024-t003:** Sociodemographic characteristics of sample (*N* = 500).

Sociodemographic Characteristics	*f* (%)	Sociodemographic Characteristics	*f* (%)
Place of residence		Occupational Status	
• In town	328 (65.6)	• Students	306 (61.2)
• In countryside	172 (34.4)	• Working students	88 (17.6)
Civil Status		• Employed	81 (16.2)
• Single	244 (48.8)	• Unemployed	25 (5.0)
• In a relationship	256 (51.2)		
Educational Level		Type of student course of study	
• Elementary school diploma	1 (0.2)	• Humanistic area	233 (60.5)
• Secondary school diploma	10 (2.2)	• Scientific area	152 (39.5)
• High School diploma	315 (63.0)		
• Bachelor’s degree	121 (24.2)	Having Faith	
• Master’s degree	44 (8.8)	• Yes	200 (40.0)
• Postgraduate training	9 (1.8)	• No	300 (60.0)
Political Orientation		Being a member of peace associations	
• Right	24 (4.8)	• Yes	36 (7.2)
• Centre	56 (11.2)	• No	464 (92.8)
• Left	266 (53.2)		
• No interest in politics	154 (30.8)		

**Table 4 ejihpe-15-00024-t004:** Item descriptive statistics and confirmatory factor analysis (*N* = 500).

Items and Dimensions	Descriptive Analysis				CFA	
	*M*	*SD*	*Sk*	*K*	*λ*	*R* ^2^
Item 1	3.83	1.04	−0.679	−0.185	0.844	0.539
Item 2	3.61	1.11	−0.469	−0.594	0.799	0.711
Item 3	4.09	1.05	−1.079	0.486	0.720	0.638
Item 4	2.70	1.09	0.203	−0.656	0.748	0.645
Item 5	3.81	1.12	−0.775	−0.135	0.706	0.653
Item 6	3.52	1.16	−0.296	−0.890	0.734	0.519
Item 7	3.07	1.14	0.030	−1.048	0.803	0.559
Item 8	3.15	1.15	−0.023	−1−046	0.808	0.499
Item 9	3.21	1.26	−0.112	−1.072	0.671	0.450
Item 10	3.63	1.25	−0.524	−0.819	0.695	0.482
WWP	3.61	0.88	−0.628	−0.111	0.813	0.661
WWF	3.32	0.98	−0.157	−0.767	0.839	0.704
WW	3.46	0.83	−0.422	−0.237		

**Note**. *M* = mean; *SD* = standard deviation; *Sk* = skewness; *K* = kurtosis; CFA = confirmatory factor analysis. In CFA columns, absolute values of standardized factor loading (|*λ*|) are reported. *λ* = factor loading onto the specific factor. All *λ* are statistically significant with *p* < 0.001. *R*^2^ = variance explained.

**Table 5 ejihpe-15-00024-t005:** Model comparison for measurement invariance.

Model	SB*χ*^2^ (*df*)	CFI	RMSEA	Comparison	DIFF*χ*^2^ (*df*)	|ΔCFI|	|ΔRMSEA|
Configural invariance	131.455 (68) *	0.971	0.060	-	-	-	-
Metric invariance	143.548 (76) *	0.969	0.060	Configural vs. metric	11.085 (8)	0.002	0.000
Scalar invariance	169.545 (84) *	0.960	0.064	Metric vs. scalar	24.995 *	0.009	0.004

**Notes:** SB*χ*^2^ = Satorra–Bentler scaled chi-squared test; *df* = degrees of freedom; Δ = differences between indices; CFI = comparative fit index; RMSEA = root mean square error of approximation; * = *p* < 0.01.

**Table 6 ejihpe-15-00024-t006:** AVE values of WWS dimensions.

Variables	AVE	*R* ^2^	
		1	2
1. WWP	0.76	-	
2. WWF	0.75	0.26	-

**Note**. WWP = Worry about the Present; WWF = Worry about the Future; AVE = average variance extracted; *R*^2^ = squared correlations.

**Table 7 ejihpe-15-00024-t007:** Pearson correlations for convergent and discriminant validity of WWS (*N* = 500).

	1	2	3	4	5	6	7	8	9
**1. WW**	-								
**2. WWP**	0.88 *	-							
**3. WWF**	0.91 *	0.60 *	-						
**4. FOW**	0.69 *	0.56 *	0.68 *	-					
**5. EXP**	0.73 *	0.66 *	0.65 *	0.77 *	-				
**6. PHI**	0.52 *	0.37 *	0.55 *	0.91 *	0.47 *	-			
**7. RES**	0.04	0.04	0.03	−0.06	−0.06	−0.05	-		
**8. DFS**	0.30 *	0.31 *	0.23 *	0.35	0.35 *	0.24 *	−0.39 *	-	
**9. DASS**	0.25 *	0.23 *	0.21 *	0.25 *	0.25 *	0.30 *	−0.33 *	0.49 *	-

**Note:** WW = Worry about War; WWP = Worry about the Present; WWF = Worry about the Future; FOW = Fear of War; EXP = Experiential dimension of fear; PHI = Physiological dimension of fear; RES = Resilience; DFS = Dark Future Scale; DASS = Depression, Anxiety, and Stress Scale. * *p* < 0.01.

## Data Availability

The data that support the findings of this study are available from the corresponding author, upon reasonable request.

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
