# Peer review of "Development and Validation of the War Worry Scale (WWS) in a Sample of Italian Young Adults: An Instrument to Assess Worry About War in Non-War-Torn Environments"

_ejihpe, 2025, doi:10.3390/ejihpe15020024_

Round 1

Reviewer 1 Report

Comments and Suggestions for Authors

The manuscript is well structured and significantly contributes to the field of psychological assessment through the development of the War Worry Scale (WWS). The validation process of the scale is thorough, and the discussion effectively interprets the results. Some suggestions for improving the manuscript are provided below:

1.     It would be interesting to discuss how parenting styles may influence young adults' preoccupation with war and their tolerance for uncertainty. In particular, several studies show that helicopter parenting, characterized by overprotection and control, may increase emerging adults' anxiety and intolerance of uncertainty. Incorporating this perspective could deepen understanding of individual differences in war-related anxiety. In this regard, a valuable reference for this discussion may be this recent scoping review on helicopter parenting and psychological adjustment in college students (DOI: 10.1080/00221325.2024.2413490), which highlights how overprotective parenting may contribute to anxiety and difficulty coping with uncertainty, both of which are relevant to war-related concerns.

2.     As the manuscript discusses the vulnerability of young adults to macro social stressors, it would be helpful to include a discussion of how the COVID-19 pandemic affected the psychological well-being of this population to strengthen the rationale for developing the War Worry Scale. Including such a discussion would provide valuable insights into how the psychological well-being of young adults was affected by the pandemic.

3.     The introduction mentions the influence of the media on war-related distress. It would be helpful to explore how media exposure interacts with individual characteristics to increase psychological distress.

4.     The manuscript could benefit from a short section on how the WWS could be used in clinical or educational settings to identify at-risk individuals and inform interventions.

5.     When reporting psychometric properties, ensure consistency in presenting fit indices (e.g., always report CFI, TLI, RMSEA, and SRMR with their thresholds).

Comments on the Quality of English Language

Minor grammatical errors have been found throughout the manuscript. Careful proofreading is recommended to improve clarity.

Author Response

Reviewer 1

Comments and Suggestions for Authors

The manuscript is well structured and significantly contributes to the field of psychological assessment through the development of the War Worry Scale (WWS). The validation process of the scale is thorough, and the discussion effectively interprets the results. Some suggestions for improving the manuscript are provided below:

- Thank you very much for your positive feedback on our work.

  1. It would be interesting to discuss how parenting styles may influence young adults' preoccupation with war and their tolerance for uncertainty. In particular, several studies show that helicopter parenting, characterized by overprotection and control, may increase emerging adults' anxiety and intolerance of uncertainty. Incorporating this perspective could deepen understanding of individual differences in war-related anxiety. In this regard, a valuable reference for this discussion may be this recent scoping review on helicopter parenting and psychological adjustment in college students (DOI: 10.1080/00221325.2024.2413490), which highlights how overprotective parenting may contribute to anxiety and difficulty coping with uncertainty, both of which are relevant to war-related concerns.

-  We thank the reviewer for this comment. The role of the parents could be a crucial factor of worry about the war and we have included it as a future perspective.

  1. As the manuscript discusses the vulnerability of young adults to macro social stressors, it would be helpful to include a discussion of how the COVID-19 pandemic affected the psychological well-being of this population to strengthen the rationale for developing the War Worry Scale. Including such a discussion would provide valuable insights into how the psychological well-being of young adults was affected by the pandemic.

-  Thank you for the suggestion. We have emphasized the impact of the pandemic on young people’s mental health more strongly in the introduction, highlighting its contribution to increasing the psychological vulnerability of the youth demographic to new collective traumatic issues.

  1. The introduction mentions the influence of the media on war-related distress. It would be helpful to explore how media exposure interacts with individual characteristics to increase psychological distress.

- Thank you for your suggestion, it was enlightening. We have enriched the discussion on the relationship between media exposure and mental health in the introduction, also to better highlight the importance of discussing a topic such as war in a context not directly involved.

  1. The manuscript could benefit from a short section on how the WWS could be used in clinical or educational settings to identify at-risk individuals and inform interventions.

- We have further enriched the conclusion section with the suggestions provided. Thank you.

  1. When reporting psychometric properties, ensure consistency in presenting fit indices (e.g., always report CFI, TLI, RMSEA, and SRMR with their thresholds).

- Dear Reviewer, thank you for this comment. We have always included indices and cutoffs, as suggested.

Comments on the Quality of English Language

Minor grammatical errors have been found throughout the manuscript. Careful proofreading is recommended to improve clarity.

- We have carefully reviewed the English throughout the manuscript. The changes made are highlighted in blue. Thank you for the feedback.

Reviewer 2 Report

Comments and Suggestions for Authors

Thank you for the opportunity to review this work. It deals with an insteresting and relevant topic. The article is well-written, grounded in the literature and all the analyses are really well detailed.

The only conceptual issue that I see is the following: authors state that there is no other tool to assess worry of war; however, there are some instruments that measure war anxiety, war-related stress, etc. It is true that the definition of worry has been presented; nontheless, I think it would be useful to provide more extensive arguments on the specific contributions of this instrument, on this particular construct, rather than the other ones; and the specificities of this construct.

As minor issues:

- The most current view on validity tend to consider it as a unitary concept: there is no such think as "content validity", "construct validity", "concurrent validity", etc.; rather, there is validity evidence based on content, construct, etc.

- I found very interesting the first inductive approach; however, I miss some details about what was asked to the 200 young adults, and how were the narratives collected.

Comments on the Quality of English Language

I believe that some sentences would benefit from a language revision. For example: "I am worried about the outbreak of World War III or nuclear" (line 244): it would be "World War III or a nuclear war"?

Author Response

Reviewer 2

Thank you for the opportunity to review this work. It deals with an insteresting and relevant topic. The article is well-written, grounded in the literature and all the analyses are really well detailed.

- Thank you for your positive feedback on our work

The only conceptual issue that I see is the following: authors state that there is no other tool to assess worry of war; however, there are some instruments that measure war anxiety, war-related stress, etc. It is true that the definition of worry has been presented; nontheless, I think it would be useful to provide more extensive arguments on the specific contributions of this instrument, on this particular construct, rather than the other ones; and the specificities of this construct.

- Thank you for your suggestion, which we consider highly relevant. We have clarified in the text (introduction paragraph) that the instruments mentioned are not adapted to the cultural context of the present study, an aspect that was not adequately addressed before. At the same time, we have expanded the introduction paragraph to better explain the motivations behind our choice of the worry construct, rather than anxiety or fear. The latter mental processes are often associated with proximal and immediate stressors, whereas worry, by activating a more enduring reflective dimension, may be a more useful construct for detecting the psychological impact of war in Italy. We have included this addition in the text.

As minor issues:

- The most current view on validity tend to consider it as a unitary concept: there is no such think as "content validity", "construct validity", "concurrent validity", etc.; rather, there is validity evidence based on content, construct, etc.

- Thank you for allowing us to specify the different approaches to the study of validity, which we have included in the data analysis section.

- I found very interesting the first inductive approach; however, I miss some details about what was asked to the 200 young adults, and how were the narratives collected.

- Thank you for your suggestion, it allowed us to provide more information on the inductive approach, which was fundamental for the construction of the WWS. We have included more detailed information in paragraph 2.1.

Comments on the Quality of English Language

I believe that some sentences would benefit from a language revision. For example: "I am worried about the outbreak of World War III or nuclear" (line 244): it would be "World War III or a nuclear war"?

- Thank you for pointing out this inaccuracy, we have corrected the translation in the text. Finally, we revised the English of the entire manuscript and the changes made are highlighted in blue. Thank you.